# The Potential of Wood Vinegar to Replace Antimicrobials Used in Animal Husbandry—A Review

**DOI:** 10.3390/ani14030381

**Published:** 2024-01-25

**Authors:** Gil Sander Próspero Gama, Alexandre Santos Pimenta, Francisco Marlon Carneiro Feijó, Tatiane Kelly Barbosa de Azevedo, Rafael Rodolfo de Melo, Gabriel Siqueira de Andrade

**Affiliations:** 1Graduate Program in Forest Sciences, Forest Engineering, Universidade Federal do Rio Grande do Norte, Rodovia RN 160, km 03 s/n, Distrito de Jundiaí, Macaíba CEP 59.280-000, Brazil; gilsander.pgama@gmail.com (G.S.P.G.); alexandre.pimenta@ufrn.br (A.S.P.); tatianekellyengenheira@hotmail.com (T.K.B.d.A.); gabrielsiqueira96@gmail.com (G.S.d.A.); 2Graduate Program in Environment, Technology, and Society—PPGATS, Laboratory of Veterinary Microbiology and Laboratory of Wood Technology, Universidade Federal Rural do Semiárido—UFERSA, Av. Francisco Mota, 572—Bairro Costa e Silva, Mossoró CEP 59.625-900, Brazil; rafael.melo@ufersa.edu.br

**Keywords:** wood vinegar, pyroligneous acid, animal husbandry, conventional antimicrobials, alternative antimicrobials

## Abstract

**Simple Summary:**

One of the most significant challenges nowadays in animal husbandry is the replacement of conventional antimicrobials as growth promoters. Reports about multi-resistant bacteria and environmental contamination caused by antimicrobials increasingly corroborate it. However, any new product developed for such must bring about high performance of animals and economic return for the farmers so the production chain remains profitable. In this context, many research works have demonstrated the potential of wood vinegar from the carbonization process as an efficient antimicrobial agent for animal husbandry. Wood vinegar is a natural and renewable product and may be a valuable alternative to conventional antimicrobials if adequately assessed and directed. In the present work, the potential of wood vinegar as an antimicrobial for animal husbandry is highlighted through several examples of the successful use of the product in managing swine, cattle, and poultry.

**Abstract:**

The indiscriminate use of antimicrobials in animal husbandry can result in various types of environmental contamination. Part of the dose of these products is excreted, still active, in the animals’ feces and urine. These excreta are widely used as organic fertilizers, which results in contamination with antimicrobial molecules. The impacts can occur in several compartments, such as soil, groundwater, and surface watercourses. Also, contamination by antimicrobials fed or administrated to pigs, chickens, and cattle can reach the meat, milk, and other animal products, which calls into question the sustainability of using these products as part of eco-friendly practices. Therefore, a search for alternative natural products is required to replace the conventional antimicrobials currently used in animal husbandry, aiming to mitigate environmental contamination. We thus carried out a review addressing this issue, highlighting wood vinegar (WV), also known as pyroligneous acid, as an alternative antimicrobial with good potential to replace conventional products. In this regard, many studies have demonstrated that WV is a promising product. WV is a nontoxic additive widely employed in the food industry to impart a smoked flavor to foods. Studies have shown that, depending on the WV concentration, good results can be achieved using it as an antimicrobial against pathogenic bacteria and fungi and a valuable growth promoter for poultry and pigs.

## 1. Introduction

To some extent, most of the productive inputs given to animals during their breeding and veterinary treatment can cause environmental contamination. Among these inputs, antimicrobials stand out [1,2,3,4,5], prompting concern about public health and food safety [6]. One of the main impacts is the contamination of arable land, usually when animal excrement is used as organic fertilizer. Residues of antimicrobials remain in the excrement and, through incorporation in soils, spread to the nearby environment, resulting in various types of contamination [7,8,9,10,11].

Environmental contamination caused by antimicrobials has attracted great concern in recent years, mainly because studies have revealed that using them in animal management for growth promotion and disease treatment is more deleterious than human use [12]. In addition to their significant influence on the emergence of microbial resistance to antimicrobial drugs, antimicrobial residues originating from animal production also contaminate soil [13,14], water [8,15], and food products [3]. Furthermore, animal husbandry’s widespread use of antimicrobials interferes with agroecological production [1]. These substances pose noteworthy challenges to human health by increasing microbial resistance to pharmaceuticals [6], thus aggravating the occurrence of severe diseases due to consuming products contaminated with these drugs [16]. Farmers and those producing meat and dairy products face the challenge of maintaining the health of their animals and farms’ profitability, which would be more challenging without conventional antimicrobials [17,18]. Besides antimicrobials, acaricides are also employed extensively in agriculture, in this case to control ectoparasites. These agents can have the same dynamics of contamination as antimicrobials, being able to pollute soil, water, and plant and animal products [19,20,21].

Therefore, developing natural alternatives to these products to eliminate or mitigate environmental contamination is an urgent task. A product that deserves particular attention for this purpose is wood vinegar (WV). WV is a natural product originating from the thermal degradation of wood. It has an extensive range of practical applications, primarily in agriculture [22,23,24,25,26]. WV has some particularities, such as chemical composition and pH, that enable its variable application [24,27,28]. Additionally, recent innovative studies have demonstrated the potential of WV for biomedical applications due to its antioxidant and wound-healing properties [29,30]. Based on previous works with various types of wood and other lignocellulosic raw materials, many researchers have investigated the effectiveness of WV as an antimicrobial agent to compose antiseptics, among other products [25,31,32,33,34].

Although other authors have comprehensively studied environmental contamination by antimicrobials, there is a gap in the literature regarding the possible use of wood vinegar to replace conventional products. In this context, this review is focused on the environmental impacts of the antimicrobials widely employed in animal husbandry and their implications for human health. To compose this review, we searched for articles on environmental contamination using conventional antimicrobials in animal management. Also, we searched for articles on using pyroligneous extract as a natural antimicrobial based on its properties, application forms, and results of experiments reported in specialized scientific journals. The main databases (in alphabetical order) consulted were Academic Search, Analytical Abstracts, Biological Abstracts, Chemical Abstracts Service, Google Scholar, JSTOR, Medline, PubMed, ScienceDirect, and Scopus. The reading of non-open access articles was granted by the Office to Coordinate Improvement of Higher Education Personnel (CAPES), through a partnership established with Rio Grande do Norte Federal University (UFRN).

The ways WV is used as a natural antimicrobial in many applications and its real potential to replace synthetic products are described in the context of the sustainability of industrial production, from the source to final usage without environmental impacts.

## 2. Environmental Impacts of Antimicrobials

### 2.1. Soil Contamination

Antimicrobial residues are one of the primary causes of soil contamination due to animal husbandry. Veterinary antimicrobials are widely used to treat various animal diseases [2] and to promote growth [4]. Studies estimate that the use of antimicrobials in animal production is significantly greater than human use in antimicrobial therapy [12]. For example, roughly 80% of antimicrobials produced in the United States are allocated to livestock use [4]. The annual consumption of animal antimicrobials per kg (total live weight) can reach up to 172 mg [35]. This high level has caused significant environmental problems. The World Health Organization has warned about the need to stop the preventive use of these medicines in healthy animals, aiming to prevent the spread of antimicrobial resistance [6]. In addition to the significant influence on the emergence of microbial resistance, the excessive use of antimicrobials brings severe environmental problems regarding soil contamination [5,36,37].

The main form of environmental impact occurs when waste products from animal husbandry, such as feces and urine, are already contaminated with antimicrobial molecules and are employed for fertilization and irrigation of arable land [9,13,35,38]. Studies have estimated that anywhere from 40 to 90% of the antimicrobial dose used for treatments is excreted and remains active in animal excrement [39]. In an evaluation carried out in different soils in China, a wide variety of antimicrobials was observed, among which we can mention sulfadiazine, sulfamerazine, sulfamethoxazole, tetracycline, norfloxacin, ciprofloxacin, and erythromycin [13]. In Belgium, an analysis was carried out directly on fertilizer produced by mixing old and fresh manure. Several antimicrobials were detected, particularly flumequine at 536 µg per kg of manure [9]. Soils contaminated by these drugs can contain up to 432 µg of antimicrobials per kg of soil [13]. In addition to contamination from these drugs, contamination can also originate from their manufacturing facilities, which release high concentrations of these drugs into the environment [40].

Therefore, environmental contamination by antimicrobials and their residues interferes with the ecological functions of natural microbial populations [5,41]. Antimicrobial molecules can remain in soil and water for long periods, significantly modifying the dynamics of microbial populations, affecting their ability to metabolize carbon, limiting their enzymatic activity, and altering the relative abundance of microorganisms [36,42]. Agents such as chlortetracycline, tetracycline, and oxytetracycline can also change the natural enzymatic activity of the soil [37]. This poses a major environmental problem since this enzymatic activity is directly related to the cycling of nutrients and their bioavailability to indigenous plants and crops [43,44].

Even after being treated by composting and/or anaerobic digestion, animal wastes can still contain significant amounts of antimicrobials, promoting the dissemination of treated waste into the soil [45]. As reviewed by Cycón et al. [42], antimicrobials in the soil can undergo different transformation or degradation processes, such as direct hydrolysis and uptake by plants. Nevertheless, knowledge in this regard is still insufficient to ensure the safe release of antimicrobial residues in the environment. The persistence of antimicrobials in natural environments depends on several factors and varies according to each product’s nature and degradation rate [46]. One study demonstrated that when the pollutant charge was not replaced, the concentration of antimicrobials in the soil decreased from 150 to 7.6 mg/kg after 30 days. However, despite this decrease, persistent antimicrobials still negatively influence soil microbial communities [41]. Furthermore, other studies have revealed that azithromycin and ciprofloxacin, for instance, degrade very slowly in the environment and can remain in the soil for 408 to 990 and 1155 to 3466 days, respectively [47]. This high recalcitrance time is enough to cause severe disequilibria in soil microbiota dynamics and metabolism.

Another form of soil pollution caused by antimicrobial drugs is contamination by resistant bacteria resulting from the increasing presence of microbial resistance genes [14,48,49]. In general, the forms of contamination are usually the same as for other pollutants: using contaminated waste from animal husbandry for fertilization and irrigation [50]. Bacterial strains like *Escherichia coli* are commonly found in calf slurry and manure. One study revealed that 88% of *E. coli* strains found in manure and 23% of those detected in calf manure in Belgium were resistant to antimicrobials [9]. Animal urine can also contain strains with high microbial resistance, as described by Yousefi and Torkan [51]. The horizontal transfer of resistance genes further aggravates the levels of ecosystem interference [52]. Using chicken litter as an organic fertilizer can also easily spread resistant strains in the soil, further increasing the selective pressure on microorganisms [53]. Unfortunately, the techniques for remediating soil contaminated with these resistance genes in bacteria do not have high success rates [50]. According to the same authors, the increase in the proliferation of microbial resistance genes is directly related to the increased concentration of antimicrobials in the habitat. The presence of these drugs imposes selective pressure on microorganisms, causing resistant genes to stand out as an adaptive strategy. Furthermore, these types of genes persist in the environment for long periods. Agga et al. [14] reported the presence of high concentrations of these genes in animal confinement areas, even two years after the use of the area had been suspended. Studies have increasingly demonstrated the need for investigations to mitigate this contamination [54].

### 2.2. Water Contamination

In addition to severe soil contamination, antimicrobial residues are widely found in surface and underground water. As previously mentioned, the primary source of this pollution is the use of excrement and wastewater from livestock farming, aquaculture, hospitals, and antimicrobials [8,55,56]. Using and releasing this polluted water allows antimicrobial molecules and particles to reach the environment, contaminating and increasing the risk of spreading microbial resistance [5]. Furthermore, these products directly affect the biota of aquatic ecosystems such as rivers, canals, and lakes, causing fish to suffer toxic effects, oxidative stress, histopathological lesions, etc. [8,57,58]. Natural aquatic formations, rivers, and canals in areas close to pig farming in Thailand were contaminated with high concentrations of antimicrobials of the tetracycline class [59,60]. The same has been reported in several other studies, including in Brazil, highlighting the severity of the problem [61,62,63,64,65].

Concerning water contamination, China stands out as a country that is currently using a large volume of antimicrobials. Therefore, much research has been conducted to understand and identify this contamination [10,64,65,66,67]. In Bagdad, Iraq, antimicrobial particles are so common that they have been detected in water treatment plants. Many of these products remain present after water treatment to make it drinkable [68]. A global need exists to analyze and regulate antimicrobials in drinking water. However, this type of analysis is rare in most countries, so humans have been consuming them unknowingly [64]. In the same way as surface water, groundwater is also subject to these contaminants. Pollution of groundwater by antimicrobials is directly linked to antimicrobials found in surface water and soil, reaching deep water through mechanisms such as microbial degradation, infiltration, and leaching [10]. A study in Ireland evaluated 109 groundwater sources, including wells and springs, and found antimicrobial contamination in 24%, reaching concentrations of 386 ng L^−1^ [15].

Studies show that the presence of these drugs in water can pose risks to human health and increase the development of microbial resistance since they can be absorbed by aquatic plants, which are then eaten by fish, contaminating them and the humans who eat fish [69]. Contamination of groundwater is just as problematic as in soil. Microbial resistance genes are also found in water due to the deliberate use of antimicrobials [59,70]. For example, several genes confer microbial resistance have been detected in Brazilian zoo ponds, some encoding β-lactamases and other types [71]. Research has been devoted to finding methods to treat liquid media contaminated with antimicrobial residues and resistant bacteria [72]. However, these methods are expensive and not fully effective [68]. Due to this challenge, researchers have been looking for ways to reduce the use of these drugs and, at the same time, minimize environmental contamination and the emergence of microbial resistance [56]. This requires the participation of the government in initiatives to improve control and regulate the use of these drugs [10,64].

As already mentioned, antimicrobials and other sources of environmental contamination are a growing concern in agriculture. Among the pharmaceuticals that contaminate agricultural systems, antimicrobials are particularly interesting due to the increasing risk of developing resistant bacteria. Agriculture is frequently blamed for the excessive use of commercial antimicrobials without perspectives for replacing them with natural products. As commented previously, overuse of antimicrobials can promote the selection of antimicrobial-resistant genes in livestock. The problem is aggravated by the possibility of transmission of multi-resistant bacteria from animals fed with antimicrobial growth promoters to humans through meat consumption [73]. In European countries, the chance of cross-infection of humans by bacteria from animal management resulted in the prohibition of selling feed containing antimicrobials as growth promoters [74]. As pointed out in the previous item, antimicrobial resistance in agricultural systems is often associated with applying manure or manure-derived fertilizers in cropland [75]. Therefore, rational management in agriculture through the employment of natural fertilizers has the downside of possible contamination, whereby good initiatives can have harmful consequences.

Agroecological systems are a particular case of agricultural systems involving practices directed to sustainable development, with rational management of soil and crops [76]. In this sense, harnessing wastes from livestock and crop management can make agriculture more efficient and environmentally friendly [77,78,79]. Therefore, using organic fertilizers from animal management in agroecological systems is common [80,81]. However, these systems are more subject to environmental contamination by antimicrobials than other agricultural management systems. So, the usual fertilizers used in agroecological cultivation are wastes from farming areas where pigs, cattle, and chickens are the main animals produced [38,82,83]. As reported in previous topics and mentioned above, using these contaminated wastes is one of the primary forms of soil and water pollution with antimicrobial residues [35].

Scientific investigations into the effects of antimicrobial contamination in agroecological settings have increased due to the lack of knowledge and concern about the possible implications for public health in general [1]. The review by Kuppusamy et al. [35] expressed this concern by focusing on the high excretion rate of active antimicrobials in the feces and urine of animals, depending on the type of drugs used as growth promoters and health treatment. The authors reported that 90% of the dose of amoxicillin and sulfamethazine remains in animal feces. When used as organic fertilizer, these excreta disseminate these drugs into the environment [9,13]. In addition to the direct application of animal excrement, there is also the use of residues from the biological digestion of animal waste to produce biogas and energy. Fertilizer residues from this practice can also be an important source of contaminants in agroecological systems. In this regard, an analysis carried out in biodigesters confirmed the large remaining presence of antimicrobial products, mainly ciprofloxacin, even after digestion [84]. Several studies worldwide have associated agroecological management with soil and water contamination with antimicrobials and the development of microbial resistance genes, highlighting the complexity of using antimicrobials for different purposes in animal husbandry [74,85,86,87].

In turn, the environmental consequences of antimicrobial contamination in agroecological management come from depositing contaminated residues in the same place. In summary, antimicrobials in soils due to agroecological farming can have unintended adverse effects. Kuppusamy et al. [35] highlighted that leaching these residues into groundwater can increase the development of microbial resistance, cause changes in soil microbial communities, interfere with seed germination, bioaccumulate in plants, and alter the action enzymes in the soil. Nevertheless, farming based on agroecology principles seems to be the more stable path to avoid widespread conflicts in food supply amid unexpected disruptions in everyday life, such as the COVID-19 pandemic [88]. Therefore, it is necessary to look for ways to correct or mitigate the current deficiencies of this production system.

### 2.3. Foodstuff Contamination

Another worrying consequence of the intensive use of antimicrobials for health treatment and growth promotion of livestock is the contamination of food products [89]. Consuming these products with high levels of antimicrobials can pose several risks to human health. These risks go beyond simple toxicity to the human body since the ingestion of these drugs through food further increases the emergence of resistance mechanisms to antimicrobials, induces allergies, and can lead to the development of other disorders such as anaphylactic shock, aplastic anemia, cancer, immunopathology, etc. [3,16,90]. Antimicrobial residues can be found in various food products of animal and vegetable origin. One of these is milk [91,92]. Assessments of milk sold commercially in Kenya demonstrated the presence of β-lactam residues in pasteurized and unpasteurized products [93]. Another study tested 194 milk samples and found that 127 of them were contaminated with a wide variety of these residues, such as sulfaquinoxaline, sulfisoxazole, sulfadoxine, cloxacillin, trimethoprim, dicloxacillin, tylosin, sulfapyridine, and penicillin G [94]. Several other studies have corroborated this situation [95,96,97,98]. Also, antimicrobial resistance of the bacterium *Staphylococcus chromogenes* was detected in a strain isolated from sheep’s milk and cheese [99], showing that dairy products can propagate multi-resistant microorganisms.

In Brazil, the problem is similar. Research has reported these drugs’ presence as residues in milk produced in different parts of the country, such as Paraná, Bahia, Minas Gerais, Rio de Janeiro, and Pará [7]. In Minas Gerais, there is a report of this kind of contamination and the suggestion that it could be avoided by reducing the use of these products in dairy production by raising awareness among producers of the seriousness of this problem [100]. In Paraná, a study demonstrated the presence of tetracyclic drugs, β-lactams, sulfonamides, and quinolones in milk [101]. After processing, the health risk was reduced. However, in rural regions where this milk is consumed without industrial processing, the risks are more accentuated due to antimicrobial residues in unprocessed milk [102]. Most studies evaluating contamination in milk worldwide have reported that one of the main antimicrobials found is β-lactam [92]. This qualifies as a significant public health problem since it can increase the emergence of microbial resistance through the production of β-lactamases. These microbial enzymes inactivate this group of antimicrobials [103,104]. Another aggravating factor is the higher risk in developing countries due to insufficient legislation and regulations [98].

In addition to milk, other animal products can also contain antimicrobial residues as contaminants. Studies have revealed that antimicrobials can accumulate in different animal tissues used in human food [3], a pattern detected and confirmed in several countries. Chicken meat can contain ciprofloxacin, amoxicillin, penicillin, and tetracycline residues [105]. The species of antimicrobials and their concentrations vary according to the farming region and the practices adopted, as demonstrated by a study that evaluated 336 chicken meat samples in different areas of India [11]. Other research has reported the presence of these drugs in commercially sold chicken meat [106]. Likewise, beef can retain the residues of these drugs, as determined in several works [107,108,109]. The same applies to other types of animal protein, such as pork [60,110] and fish meat [69,111]. Furthermore, these antimicrobials can also accumulate in plants [112]. An experiment found a significant accumulation of ciprofloxacin and oxytetracycline in cabbage, spinach, and endives grown in contaminated soils [113]. The same happened with tomato plants exposed to soil contaminated with tetracycline and sulfamethazine [114]. Antimicrobials have a high rate of residence in different foods, and as mentioned previously, consuming these contaminated foods can promote several unwanted effects on human health [16].

## 3. Wood Vinegar—Composition, Properties, and Uses

Wood vinegar (WV) is an aqueous liquid originating from the thermal decomposition (pyrolysis) of lignocellulosic materials in the absence of oxygen or under a controlled atmosphere [24,26,115]. During carbonization or slow pyrolysis, charcoal and gaseous products (smoke) are formed [116,117]. Field measures in industrial kilns have shown that roughly 60–65% of the initial bone-dry wood turns into smoke during carbonization [118,119]. When leaving the reaction bed, the smoke from slow pyrolysis can be carried through devices where a portion condenses, generating the raw pyrolysis liquids. These liquids, after settling, separate into two parts, an aqueous one and an oily one. The aqueous fraction is the WV, and the oily one is called vegetable tar [24,120]. In turn, the oily fraction can contain a supernatant consisting of light oils [28,121], depending on the type of raw material, whether hardwood or softwood. Fast pyrolysis processes can produce aqueous liquids from carbonization with properties like WV. The yields and chemical compositions vary according to the tree compartment of origin [122]. Several lignocellulosic raw materials, such as hardwood, softwood, and bamboo, have been investigated as potential sources of WV [24,122,123]. Crop and forest wastes such as cotton stalks, softwood, and bamboo sawdust have also been investigated [124]. As an adjuvant in soil, WV was referred to as an enhancer that could improve the complexation of remediation materials and Pb(II) ions, which is an exciting application [125].

### Properties and Chemical Composition

WV is also known as pyroligneous acid or liquid smoke. The latter name is used when the product imparts a smoky flavor to meat products, sauces, and other foods [126]. WV is essentially an additive with a strong smoke odor employed worldwide in the food industry and used by some 80% of manufacturers of smoked products, providing the characteristic flavor and smell [127]. In addition to imparting a smoky flavor, WV can also be applied to preserve food (fish, meat, and sausages) by extending the shelf life [128]. In this regard, WV is a natural chemical product with properties recognized and certified by several renowned international bodies, such as the Chemical Abstracts Service (CAS 8030-97-5), Flavor and Extract Manufacturers Association (FEMA 2967), European Inventory of Existing Chemical Substances (EINECS 232-450-0), Harmonized Commodity Description and Coding Systems (HS 2915.50.5000), and Food and Drug Administration (FDA-21-CFR compliance 172.515).

WV is composed mainly of water (80–85%) and a mixture of at least 200 organic compounds [24,129,130] of different chemical classes, as follows: organic acids; alcohols; esters; furans; ketones; phenolic compounds; pyrans; and other minor compounds [27,32,131,132]. WV has an acidic character, with pH varying between 2.5 and 3.6 [133,134] and density slightly higher than water, ranging from 1.008 to 1.020 g cm^−3^ [24]. Its color varies from reddish brown or yellowish brown to dark brown [135]. WV has been used in several areas, mainly agriculture, where the product has many applications as a partial or total substitute of pesticides and fungicides [22,25]. Usually, a significant fraction of WV is composed of organic acids, phenolic compounds, and sometimes furfural, depending on the original raw material [23,24,132]. Among the varied chemical components of WV, phenolic compounds are recognized as the agents responsible for the biological effects of WV as an antimicrobial agent and for other uses [22,25,136]. All told, WV contains at least 20 phenolic compounds [24,132].

## 4. Wood Vinegar—An Effective Natural Antimicrobial Agent

Numerous studies have demonstrated the antimicrobial properties of WV against pathogenic microorganisms and its natural, safe, and eco-friendly characteristics [25,28,137,138]. Due to the chemical particularity of the WV originating from each plant species, different products have been tested as antimicrobial agents worldwide [24,28,32,139]. The results have demonstrated that WV is a valuable antimicrobial agent, encouraging its consolidation as an alternative to antimicrobials in practical applications. In in vitro tests, Harada et al. [140], using WV from bamboo (*Phyllostachys pubescens*), demonstrated inhibition of *Escherichia coli*, *Staphylococcus pseudintermedius*, and *Pseudomonas aeruginosa*. Hou et al. [141] described satisfactory results against several pathogens, including *Enterobacter aerogenes*. In Canada, using WV from a mixture of wood was active against *Salmonella enterica* and *Lactobacillus acidophilus* at concentrations ranging from 0.8 to 3.2% [142]. Other researchers have also tested the antimicrobial activity of WV from different tree species and have stressed their potential to replace conventional antimicrobials [23,25,32,129,137,143]. Some authors have reported WV’s effectiveness as an anti-inflammatory agent [144,145]. Antiviral activity of WV from hardwood, softwood, and bamboo was also reported [123,146].

Besides being a natural product from renewable sources and helping to mitigate atmospheric emissions, since WV is collected from the smoke emitted during the production of charcoal, another advantage of this product is that it poses difficulties to the development of microbial resistance. This is due to the wide variety of bioactive compounds in its chemical composition, which have synergistic antimicrobial actions. As mentioned above, phenolic compounds are recognized as responsible for WV’s biological effects when used as an antimicrobial agent. Since at least 20 phenolic compounds are usually contained in WV, the probability of microorganisms simultaneously developing resistance mechanisms against all these substances is remote [33,142]. In addition to its antibacterial action, WV has high antifungal potential. In Indonesia, WV from cocoa pod shells was found to act against fungal strains, including *Candida albicans*, with inhibition halos ranging from 6 to 6.12 mm [147]. Other studies have also reported the antifungal action of WV against *C. albicans* and other fungi [23,32,33,148].

Several mechanisms explain the antimicrobial effects of WV. An assessment of the effects of WV against *C. albicans* demonstrated the destruction of the cell wall of this yeast [149]. The presence of organic acids in its composition can cause damage to the cell walls of bacteria and fungi, changes in the normal pH and other characteristics of their cytoplasm, and alterations of the microorganisms’ genetic material [32,150]. Changes in pH can result in disequilibrium in the capacity of microorganisms to produce H+ at a regular rate, interfering negatively with the osmotic pressure of both the cytoplasmatic membrane and cell wall, which can result in their distortion and rupture [151,152]. On the other hand, phenolic compounds can thin the cell wall and cause cellular depletion and the dispersion of the ribosomes [152]. Phenolics can also promote the displacement of cytoplasmic membranes, impairing their integrity and hence causing the leakage of cytoplasm components [153].

## 5. Use of Wood Vinegar in Animal Husbandry

This item highlights the use of WV as an input in livestock management, showing the state of the practice and the results of global studies of WV from different lignocellulosic raw materials or biomass wastes. Several assessments of WV as a growth promoter in animal husbandry have shown its potential to replace conventional antimicrobials in this application. Around the world, research has demonstrated the efficiency of this product in poultry, swine, and cattle. To better illustrate the applications of WV in these fields, Table 1 summarizes the relevant information.

The results of WV addition in livestock diets are relatively scarce. There are many more experimental results of powdered charcoal and biochar on cattle performance than findings regarding WV-only use, as displayed in Table 1. Kook and Kim evaluated the effects of supplemental levels of bamboo WV on the growth performance, serum profile, and meat quality of Hanwoo cows [154] in Korea. Concentrate diets were supplemented with the product with a 3% addition level. This improved the marbling score and crude fat content, decreased the shear strength and cholesterol content, and improved the taste according to sensory evaluation. Adding biochar to cattle feed increases body weight, and using activated charcoal combined with WV has been cited as reducing cryptosporidiosis in goats and cattle. Regarding the assessment of biochar and WV mixtures, O’Reilly et al. [155] investigated the effects of different mixtures on in vitro batch ruminal culture fermentation using other feed substrates as references. Based on their experimental results, they stated that biochar is not effective for methane mitigation in ruminant livestock despite previous works that reported success in CH_4_ emission reduction.

Concerning pig farming, better growth performance and feed digestibility were found among animals with diets containing variable levels of WV. Some results are shown in Table 1. For instance, Mekbungwan et al. [156] assessed histological intestinal villus alterations in piglets that received a raw pigeon pea diet, including charcoal powder and WV. The results demonstrated that the intestinal features could be atrophied by feeding pigeon pea meal to the animals, resulting in decreased growth performance. In another experiment, Choi et al. [157] assessed the feed value of increasing levels of WV given to weanling pigs until 28 days of age against positive controls with apramycin and a negative one containing feed without any additive. In conclusion, the authors stated further research was necessary for its consolidation as a promising product to replace conventional antimicrobials in pig farming.

In turn, Chu et al. [158] demonstrated that including bamboo charcoal or bamboo WV as an alternate antimicrobial in the diet of fattening pigs led to better growth performance, immune responses, and fecal microflora populations. They measured decreased cortisol levels, higher average daily weight gain, and better feed efficiency with the feed additive. Thus, bamboo charcoal or WV has the potential as an additive in pig production instead of conventional antimicrobials since growth performance, immune response, and fecal microflora populations were improved. Additionally, Wang et al. [159] evaluated the effects of feeding bamboo vinegar combined with an acidifier (a mixture of organic acids) as an antimicrobial substitute on the growth performance and intestinal bacterial communities of weaned piglets. The results indicated that the tested additives could effectively replace antimicrobials in the diets of piglets without adverse effects on production, contributing to a higher diversity of the intestinal bacterial population compared to antimicrobials.

**Table 1 animals-14-00381-t001:** Uses of various types of WV in animal management (cattle, swine, and poultry).

Animals	Type of WV	Concentration	Frequency of Use	Effects	Author
Cattle	Bamboo	3%	Once a day	Improvement of meat quality (taste and marbling), higher contents of crude fat, less shear strength, and less cholesterol content in meat	Kook & Kim, 2003 [154]
	Nekka-rich	10 g (daily dosage)	Included in a milk surrogate for 4 days (every 8 h)	Control of *Cryptosporidium parvum* in calves	Watarai et al., 2008 [160]
	Obionekk	1.25 g (daily dosage)	Included in a milk surrogate for 14 days (every 8 h)	Control of *Cryptosporidium parvum* in goats	Parauda et al., 2011 [161]
Swine	Nekka-rich	3%	Inclusion in feed for 30 days	Improvement of feed conversion and villi height	Mekbungwan et al., 2008 [156]
	Commercial WV	0.3%	Inclusion in feed for 28 days	Improvement of digestibility and control of undesirable coliforms	Choi et al., 2009 [157]
	Bamboo	0.3%	Inclusion in feed for 42 days	Improvement of performance and stress reduction	Chu et al., 2013 [158]
	Bamboo	0.4% WV + 0.25% acidifier	Inclusion in feed for 25 days	Improvement in intestinal microbiota	Wang et al., 2013 [159]
	*Acacia auriculiformis* wood	0.3% WV or 0.2% WV + 0.8% biochar	Inclusion in feed twice a day	Control of diarrhea and sulfide hydroxide emissions	Chao et al., 2016 [162]
	Bamboo	0.5%	Inclusion in feed for 35 days	Regulation of expression levels of mRNA in immune organs	Huo et al., 2016 [163]
	*Garcinia mangostana*	0.4 to 0.8%	Inclusion in feed for 5 days	Improvement of digestibility	Rodjan et al., 2018 [164]
	Not informed	5.0%	Inclusion in feed for 4 weeks	Improvement in weight gain	Macasait et al., 2021 [165]
	*Quercus acutissima* wood	0.1%	Inclusion in feed for 16 weeks	Improvement in weight gain and total digestibility of nutrients	Sureshkumar et al., 2021 [166]
Poultry	Biochar + WV	0.5 and 1%	Inclusion in feed	Improvement of egg production and intestinal villi height, less emission of fecal ammonia	Yamauchi et al., 2010 [167]
	Silicic acid (commercial product) and bamboo vinegar	0.3%	Inclusion in feed for 112 days	Greater weight gain	Rattanavut et al., 2012 [168]
	Silicic acid (commercial product) and bamboo vinegar	0.2%	Inclusion in feed for 49 days	Improvement of intestinal villi number and height	Rattanawut & Yamauchi, 2015 [169]
	Biochar and bamboo vinegar (8 kg of powder + 3 L of BV)	1 and 1.5%	Included in daily feed	Improvement of egg quality, digestibility, and control of *Escherichia coli* and *Salmonella* sp.	Rattanawut et al., 2017 [170]
	Not informed	0.833%	Inclusion in feed (twice a day) for 84 days	Improvement of laying performance and egg quality	Nunes, 2019 [171]
	WV from *Eucalyptus urophylla × Eucalyptus grandis* (clone GG100)	2.5%	Inclusion in feed for 42 days	Improvement of body weight gain, feed conversion, and feed consumption	Diógenes et al., 2019 [172]
	WV from the hull of Spina date seed	0.2%	Inclusion in feed for 50 days	Improvement of egg yolk quality and decrease in n-6 fatty acids	Zhao et al., 2019 [173]
	EP of *Myristica fragrans* and *Acacia confuse*	0.5 or 1%	Added to water twice per day	Improvement of intestinal villi height	Hanchai et al., 2021 [174]

Nekka-Rich (Cape Cross, Jeffreys Bay, South Africa) and Obionekk (Obione, Charentay, France) are trademarks.

Chao et al. [162] evaluated using charcoal powder and WV to prevent diarrhea and reduce environmental pollution from swine production. The authors determined that the additives could reduce the incidence of diarrhea in pigs and that the concentration of hydrogen sulfide in pig houses decreased in the experimental groups that received a mixture of charcoal and WV in different proportions in the diets. The results demonstrated the undeniable positive effects of charcoal and WV in improving animal health and decreasing environmental pollution from pig breeding. Another study on preventing diarrhea in piglets was conducted by Khai et al. [175], utilizing a WV–activated charcoal mixture. They found positive results in weanling and post-weaned pigs by preventing diarrhea in the rainy and dry seasons. When assessing the effects of mangosteen WV as a potential additive to improve nutrient digestibility in growing pigs, Rodjan et al. [164] found positive results, suggesting that WV can be used as a potential additive in pig farming.

Macasait et al. [165] evaluated the growth performance of pigs and the nutritional and microbial contents of wet and fermented commercial feed containing different levels of WV. Although no differences in microbial and nutritional contents were noted in the fermented feed, a significantly higher profit was determined from pigs fed wet and fermented commercial hog feed containing 5% WV. Also, Sureshkumar et al. [166] assessed the effect of dietary inclusion of WV supplementation on growth performance, nutrient digestibility, and meat quality of grower-finisher pigs, finding enhancement not only of growth performance but also total tract digestibility of nutrients with no effects on lean meat percentage and backfat thickness, claimed to be exciting results.

In an interesting approach, the efficiency of WV was assessed to control offensive odors from piggery wastes [176]. The odorants from piggery wastes were identified as ammonia, methyl sulfide, hydrogen sulfide, butyric acid, and valeric acid. In a laboratory experiment using an air-tight vessel, the WV concentration required for deodorization was 6.6%, with removal efficiency from 70 to 90%. In situ tests at a pig farm showed that the efficiency in removing odorants was similar to that of the laboratory experiments. Besides that, flies were rarely observed, indicating that WV may play a significant role as a repellent.

In poultry farming (see Table 1), the results of experiments have demonstrated the efficiency of using charcoal powder and WV as feed additives in laying hen breeding, resulting in increases in egg production and eggshell thickness and decrease in fecal ammonia concentration and damaged egg rate [167]. Furthermore, the use of WV in feeding these animals increased weight gain of the animals and maximized cell mitoses, resulting in better intestinal health and egg quality and reduced pathogenic populations of *Escherichia coli* and *Salmonella* sp. [168,169,170]. Furthermore, Watarai and Tana [177] assessed the protective efficacy of activated charcoal containing WV (Nekka-Rich) against intestinal infection caused by *Salmonella enterica* serovar *Enteritidis* in domestic fowls. In a study by Watarai and Tana [177], the adsorption effectiveness of Nekka-Rich against *S. Enteritidis* and normal bacterial flora in the intestine, *Enterococcus faecium*, was evaluated. According to the authors, significantly less fecal excretion of *S. Enteritidis* was observed in chickens fed Nekka-Rich for ten days after the challenge. On day 15 after the challenge, *S. Enteritidis* was no longer isolated from fecal samples. On the other hand, immunization of chickens with the *S. Enteritidis* vaccine did not fully inhibit bacterial growth. Therefore, the experimental results indicated that Nekka-Rich may effectively eliminate the presence of *S. Enteritidis* in domestic fowls.

Also, Sittiya et al. [178] evaluated the effects of a wood charcoal powder and wood vinegar solution on *Escherichia coli*, ammoniacal nitrogen, vitamin C, and the productive performance of laying hens. They found that at 72 weeks of age, the performance parameters showed the highest values in the mixture dosage of 2%, while the yolk color and Haugh units were the highest in the 3% dietary group. According to the authors, the fecal *E. coli* concentration increased at the early stage of 66 weeks in the 1% mixture group without any change in ammoniacal nitrogen. They attributed the results to the reduced power of the mixture itself and the vitamin C increase through its action. These results were found with 2 to 3% supplementation levels, which were considered most suitable for tropical climates. In another study, Hanchai et al. [174] found no effect of pure WV supplemented in drinking water on growth performance, intestinal morphology, and gut microorganisms of broilers but determined improvement in villi number and height.

Supplemental feeding of laying hens with varied concentrations of WV from spina date seeds was assessed by Zhao et al. [173], aiming to decrease the n-6 to n-3 fatty acids ratio. In an experiment with broiler quails (*Coturnix coturnix*) fed with increasing concentrations of WV, Diógenes et al. [172] observed increased weight gain and feed consumption and decreased feed conversion as the additive concentration increased. Another experiment with laying quails was carried out by supplementing WV in the diet and assessing the animals’ performance and egg quality according to increasing additive levels [171]. The experiment also employed control groups where animals received feed containing conventional additives such as enramycin, probiotics, prebiotics, and essential oils. The results showed that 1.2% WV could replace the antimicrobial, probiotics, prebiotics, and plant-essential oil additives with statistically significant effects on the parameters of feed consumption, laying rate, average egg weight, egg mass, feed conversion per egg mass and dozen eggs, the relative and absolute weight of yolk, albumen, and shell, in addition to the Haugh units. Among these parameters, feed consumption and conversion were better than those determined for the negative and positive groups.

## 6. Wood Vinegar Refining

WV has been used mainly in agriculture, where the product has many applications as a partial or total substitute for pesticides and fungicides [22,25]. Among many other uses, WV from the carbonization of different types of woody biomass is effectively used worldwide as a safe food additive for human consumption. As cited previously, international regulations approve this type of use [126,127]. International standards require refining so that the WV reaches food and pharmaceutical grade, depending on the product type generated with this input. WV must undergo purification processes to reach the appropriate quality due to contaminants in its chemical composition. The acute toxicity and genotoxicity of chemical components of liquids and tars from wood pyrolysis and other lignocellulosic raw materials are comprehensively presented in the literature [179,180,181,182]. Therefore, depending on the targeted application of WV, it must be subjected to refining processes to remove contaminants. Most likely, raw WV’s main class of harmful compounds is polycyclic aromatic hydrocarbons (PAHs) [181,183]. PAHs are proven carcinogens and are classified as primary pollutants by international health agencies [122].

In the refining phase, PAHs and other contaminants are removed, along with tar residues and remaining soluble tar [180,181,184,185]. Several processes are recommended to purify WV, with decantation and filtration being the most common. Other options for refining include combining filtration with ultrafiltration, centrifugation, and distillation. However, depending on the refining method, traces of tar may remain in the final product, carrying contaminants in their structure. Vacuum refining is considered the most reliable and efficient process to refine WV and pyrolysis oils for food uses such as liquid smoke, generating a high-quality final product [120,122,132,186,187]. Even for agricultural applications, refining WV is recommended to remove soluble and insoluble tar [120]. The WV must undergo more pronounced refinement for more restrictive applications, such as research in health areas. The raw product must be subjected to a single distillation process to ensure its purity [23,32,188].

After refining, the WV is usually investigated to determine the main components, especially those with greater biological effects according to the final use. When studying the chemical composition of WV from Eucalyptus wood, 63 main components were found [132], with the methoxyphenol group having the highest representation. The most abundant components were 2,6-dimethoxy-phenol (syringol), 1,2,3-trimethoxybenzene, 2-methoxy-4-methylphenol, guaiacol, and 5-tert-butylpyrogallol. Yang et al. [28] reported the same chemical profile for WV from *Litchi chinensis* wood. Other studies have identified other compounds in WV from different forest species [24,139,142,189]. The phenolic compounds, for instance, occur in all types of WV, with concentrations that depend on the woody material from which the product originated.

Several other researchers have determined similar profiles, unsurprising since most phenolic compounds come from lignin thermal decomposition [24,115,181]. Lignin is a polymer in the chemical composition of all types of woody biomass. Therefore, WV from most of these wood sources has similar biological properties. This characteristic of various chemical compounds in WV explains its wide applications, such as in human and veterinary medicine, animal husbandry, agriculture, sustainable development, etc. [22,26,32,138,143,190]. Other contaminants, such as the heavy metals lead, cadmium, arsenic, and mercury, are also removed from WV by refining [120]. Figure 1 depicts WV cleanliness after sequential refining steps. After one month of settling, the raw product has a significant concentration of wood tar and heavy oils with which PAHs are associated [180,181]. After refining, the contaminants are entirely removed, and the final product reaches high crystallinity and suitable refractive index.

## 7. Wood Vinegar Toxicity

As previously mentioned, WV is classified as a smoky flavor additive used worldwide in the food industry, indicating that humans can ingest the product in adequate amounts without harmful effects [126,136]. For instance, Imamura and Watanabe [191] patented a bamboo WV from *Phyllostachys heterocycla*. They demonstrated that the product was safe (free of contaminants and impurities) after refining and could be ingested as a food additive in the long term. Also, applying WV as a preservative for food products dates back hundreds of years [192,193,194]. No risk to human health has been reported if the product is properly prepared and purified. So, if the product is safe for humans, it is also considered safe for animals. In general, toxicity tests of WV have been carried out based on doses given to animals via feed or water, in addition to tests related to the topical use of this product [143,172,174]. Unlike the case of antimicrobials, we found no studies determining whether WV residues are excreted in animal urine and feces.

Some research of WV’s effects on soil microorganisms and plant uptake has been carried out, indicating no harmful effects when residues of the product are incorporated in arable soils. According to Akley et al. [195], foliar application of WV on cowpea plants combined with soil drenching improved soil health, nodulation, and cowpea yields and enhanced profitability in Ghana. Koç et al. [196] demonstrated that WV obtained from hazelnut shells in concentrations up to 3% positively affected the number of soil bacteria and beta-glucosidase enzyme activity. In brief, at adequate levels, WV has various benefits in soil, such as promoting plant growth, functioning as a nematicide, increasing microbial diversity, assisting the degradation of glyphosate, and reducing the abundance of microbial resistance genes in rhizosphere soil [197,198,199,200,201,202].

Studies of WV toxicity directly related to human uses are scarce. Nevertheless, some research has revealed the safety of using this product in specific concentrations. An in vitro cytotoxicity test for developing a chitosan biofilm and WV for dental treatments demonstrated no cell cytotoxicity at 1500 and 7500 μg mL^−1^ [203]. In another in vitro study, toxicity of palm kernel WV to human skin cells was found [29]. In this work, the authors divided WV into different fractions and assessed their cytotoxicity on human skin fibroblasts. Cytotoxicity to cells was observed at a concentration of 50 μg mL^−1^ of WV, resulting in cell death. However, no cytotoxicity was observed at concentrations lower than 25 μg mL^−1^. Furthermore, at 12.5 μg mL^−1^, cell viability was significantly increased. In addition to this research, a cytotoxicity study was carried out using WV against kidney cells from neonatal hamsters. The results demonstrated toxicity because, at 10% WV, only 49.23% of the cells remained alive [204].

WV toxicity has been evaluated against the fish *Danio rerio*. Two concentrations were assessed, 2 and 5%. At 5%, WV was strongly toxic to fish, causing total mortality after exposure for 24 h, while at 2%, no mortality was found [205]. Another work [189] evaluated the acute toxicity of eucalyptus WV to the brine shrimp *Artemia salina*. Median lethal doses (LD_50_) equal to 272 and 298 mg mL^−1^ were determined for the raw and refined WV, respectively. Regarding topical use, the cytotoxicity of 20% WV incorporated into glycerin as an antiseptic product was applied on dairy goats after milking [31]. The study demonstrated that even at 20%, WV did not cause cytotoxicity in the mammary gland cells of exposed animals, revealing its safety for this purpose. These findings corroborate those presented by Soares et al. [143]. When used for 28 days to disinfect the teats of dairy cows, WV at 1% was also safe, not resulting in toxicity to the cells of the udders [34]. The results of using WV to compose antiseptics for animal health treatment strongly indicate the product’s potential to replace antimicrobials. In this regard, a study demonstrated the effectiveness of WV from *Mimosa tenuiflora* wood at a concentration of 20% as a postoperative antiseptic on female cats subjected to ovariosalpingohysterectomy [138]. In this clinical case, the proliferation of microorganisms was inhibited by WV, and the healing of surgical wounds was enhanced compared to the group treated with chlorhexidine.

It is important to mention that the data cited above are illustrative, giving a panoramic view of the absence of worrisome WV toxicity to the environment and living organisms. There are no red flags related to WV use and environmental contamination. Moreover, the product is a safe food additive when suitably refined. Besides being tested in different organisms and contexts, the variability in the WV chemical composition according to the woody raw material is an important issue to explain the variability of experimental results. Hence, dosages and concentrations that can be toxic to a particular organism will vary with WV quality and origin. Since WV is a chemical, it is necessary to control the amounts of the product released into the environment since it can be toxic to animals at excessive concentrations. Therefore, despite the safety found for WV use as a natural, eco-friendly, and biodegradable substance, further research is needed to establish the safe residue levels of the product in the environment and its possible influence on public health.

## 8. The Potential of Using WV on a Large Scale in Animal Husbandry

As noted here and comprehensively investigated elsewhere in the literature, environmental contamination by antimicrobial residues from animal husbandry cannot be downplayed or overlooked. Soil, water, and foodstuffs are subject to contamination. Therefore, researchers, health authorities, and other agents seek alternatives to prevent or mitigate the potentially harmful effects on public health caused by this usage. As displayed in Table 1 and discussed throughout the text, there are extensive variations in the experimental results of WV use in animal husbandry. These variations are associated with the different animals and the high variability of WV, which depends on the raw material and carbonization parameters.

One severe shortcoming identified in most works cited here is the lack of information about the chemical profile of the types of WV used to perform the assessments. There is no description of properties (density, color, pH, refractive index, water and acetic acid content, titratable acidity, phenolics content, and so on). Some works do not even identify the woody biomass from which the WV was produced. Information on the types of carbonization kilns and other thermal processing parameters is also lacking. Most importantly, the presence or absence of contaminants in WV is generally not addressed. Thus, many questions remain unanswered, such as whether or not the WV employed by other researchers was refined before use. This lack of information is problematic because it makes it harder to establish firm comparisons among experimental results and relate them with WV composition and properties. Still, the results described here indicate WV’s potential application in poultry and pig farming. But this will only be possible on a larger scale if WV properties and quality can be standardized and kept constant so the product can be administered to animals just like conventional growth-promoting antimicrobials.

Given the results reported here, it is appropriate to wonder why WV can have so many positive economic, environmental, and biological effects on animal husbandry and still not be widely employed. However, despite the success of WV in numerous situations, several disadvantages can be associated with the product, which makes it extremely difficult to standardize recipes and define dosages, as follows:Each carbonized wood or woody biomass produces a different type of WV; therefore, the product dosages for a particular animal can often differ broadly in efficiency as a result of variations in chemical composition. This brings uncertainty to general results, preventing standardized recommendations for use like those prevailing for conventional antimicrobials. In other words, a given antimicrobial always has the same chemical structure everywhere, so its prescription and use are easy to accomplish.WV is usually locally produced by small farmers according to traditional practices that are not always appropriate. Thus, there is no standardization of final product quality or properties. Virtually everywhere in the world, WV is produced locally in low-technology kilns in which the carbonization parameters cannot be accurately determined, generating liquid products of doubtful quality without reproducibility.One prosaic mistake many WV producers make is recovering the pyrolysis liquids from the outset of the carbonization process. Since every woody biomass has some amount of water embedded in its tissues, if the recovery starts before this moisture evaporates at the beginning of carbonization, it will remain in the final product, acting as a dilutant and decreasing the content of active compounds.Each kiln model and carbonization routine will produce WV with different yields and quality [115]. Hence, a minimum level of reproducibility is required to achieve desirable characteristics for WV from each wood or woody biomass type.No matter how efficient the carbonization kilns, their recovery accessories, and the process parameters are, the raw WV will contain polycyclic aromatic hydrocarbons that are highly carcinogenic. To achieve a high-quality final product, the raw WV must be refined.There are no legal regulations regarding WV quality. Only informal rules of thumb are typically considered. Thus, the products are sold freely regardless of their sound quality.

Therefore, besides being widely recognized as an excellent natural pesticide for agricultural uses, WV positively affects the growth of pigs, cattle, and poultry and has antimicrobial, antifungal, antiviral, anti-inflammatory, and immunomodulatory properties. However, only standardization in all phases of WV production can generate a product that can be recommended in precise dosages as conventional antimicrobials used for these same purposes. That standardization must occur from planting the raw material through the type of kiln, carbonization process parameters, and processing techniques so that the WV has constant quality and properties.

First, the global consumption of conventional antimicrobials should be determined to know if it is possible to produce WV in amounts sufficient to have a relevant share in the market for animal growth promoters. Antimicrobial sales in 41 countries for poultry, cattle, and swine production were 93,309 metric tons in 2017, a figure that is projected to reach 105,000 tons by 2030 [206]. These numbers need to be considered regarding initiatives for industrial-scale production of WV. Competition with traditional antimicrobials will not be easy due to their undeniable benefits on animal growth, high reliability, consistent quality, wide availability, and competitive prices. Successful commercial production of WV requires achieving the same qualities as conventional products. On the other hand, the increasing restrictions applied to antimicrobials as growth promoters in several countries and the prohibition of importing meat and other foods derived from animals fed with them [206] constitute a real opportunity to insert WV into the world market as an input for animal husbandry.

Therefore, to produce WV to compete in this market, some formal production standards must be determined to ensure consistent quality. For example, Brazil, the largest charcoal maker in the world, produces 4.5 metric tons annually [207]. Roughly 95% of this charcoal is consumed for metallurgy. In Brazil’s “green metallurgy”, 70% of the pig iron, 40% of the steel, 100% of silicon, and 100% of iron alloys are produced with charcoal as a reducer. Another important characteristic is that over 95% of the charcoal consumed by the metallurgical industries comes from the wood of forests planted with various eucalyptus clones, meaning all the trees are genetically equal. Combining this information with the fact the quality of eucalyptus WV does not vary broadly from one clonal wood to another [24,32], this means that some Brazilian companies could recover the liquid products from carbonization with sufficient capacity to supply large amounts of standardized WV to the world market. Based on the historical average yields of solid and liquid products from the eucalyptus wood carbonization (33 and 35%, respectively), Brazil can produce at least 3.5–4.0 million metric tons of WV annually. For each country or business group that produces charcoal, calculations can be performed to assess the capacity of WV to supply the needs of animal husbandry instead of antimicrobial use. Nevertheless, the productive chain will only be entirely renewable if the charcoal is made from planted forests due to the absence of deforestation.

To illustrate how it is possible to establish a charcoal WV chain of production that is entirely sustainable, Figure 1 depicts the production steps of Brazil’s largest pig iron and steel producers. Initially, seven-year clonal eucalyptus forests (1) provide the firewood (2), which is taken to carbonization kilns (3) and processed. The resulting charcoal (4) goes to the blast furnaces (5), where pig iron (6) is produced and exported or converted into steel (7). Recovery apparatuses (8) collect the WV. After collection, WV is transferred to settling tanks (9). After 3–4 months, the wood tar at the bottom of the tanks is separated, and the aqueous phase is filtered (10), yielding the raw WV (11), which goes to agricultural uses (12) as a pesticide, among other applications. As proposed here, a new line of products can be generated by refining (13) to remove contaminants from the raw product to obtain the refined WV (14), which would be used to formulate new products for animal husbandry, veterinary, and other uses (15).

The production chain displayed in Figure 1 can provide large amounts of WV for new applications. This production model can be attractive, especially due to the growing demand for agribusiness.

## 9. Conclusions

Antimicrobial growth promoters are practical products necessary for large-scale animal husbandry to be economically feasible. However, as depicted throughout this review, animal management antimicrobials can cause various environmental problems, including soil and water quality degradation, ecosystem changes, and accumulation in plant and animal tissues. In this sense, and considering all the experimental results summarized in this review, WV is a product with real potential to replace conventional antimicrobials since it has low toxicity as a food additive and would not cause environmental problems. The carbonization kilns, machines, and processes to make WV are well-established, and the quality parameters that make it commercially pure and contaminant-free are known. Thus, expanding the use of WV as an antimicrobial in animal husbandry depends on a formal production chain that can provide the product in large amounts with constant high quality.

## Figures and Tables

**Figure 1 animals-14-00381-f001:**
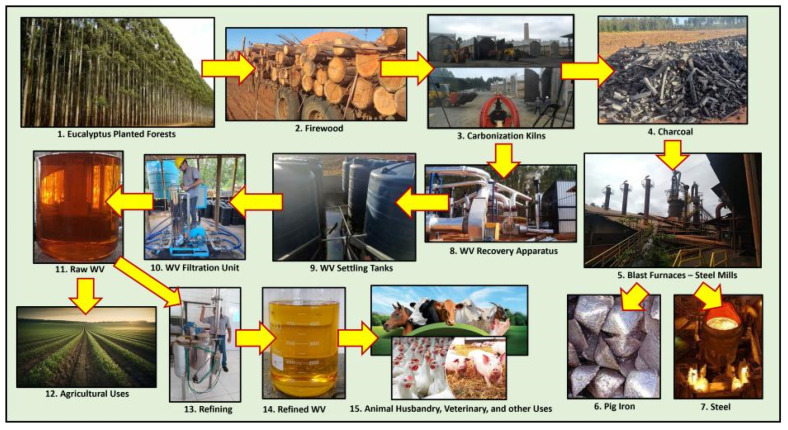
Charcoal and WV production chain of a pig iron and steel producer in Brazil with an additional step of WV refining.

## Data Availability

Not applicable.

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
