# Peer review of "The Potential of Wood Vinegar to Replace Antimicrobials Used in Animal Husbandry—A Review"

_animals, 2024, doi:10.3390/ani14030381_

Round 1

Reviewer 1 Report

Comments and Suggestions for Authors

The topic of the review is interesting. The authors write and organize the review well. However, the authors can still improve the quality of the review.

1.     Line 32 Wood vinegar (WV) should be moved to line 30.

2.     According to the second part (Environmental impact on antimicrobials), the authors have divided the subtopics into soil, water, agriculture, and foodstuff. To make it easier to follow, the author should divide or add the subtopics in the third part (properties) into soil, water, agriculture, and foodstuff.

3.     The authors should add the mechanism of action of WV against bacteria.  

Comments on the Quality of English Language

Please check some typos.

Author Response

Please, see our answers in the attached document.

Reviewer 2 Report

Comments and Suggestions for Authors

The MS ID animals-2707102 Wood Vinegar from the Carbonization Process and Its Potential to Prevent Environmental Contamination Caused by Antimicrobials from Animal Husbandry—A Review” is well written but lacks many scientific aspects. As it stands, the article reads like a list of loosely-related snippets of information in form of a book chapter, but the manuscript needs to be more directive – what is important (and why) and what is trivial. This will massively improve the article. At present form I feels it’s a book chapter lacking Tables, graphical presentation, figures etc.

Indeed, the subject is interested to focus but I do not see any novelty of the study. The aim of the study "The goal of this paper is to do a thorough review how they are made by the authors. What kind of search engine used and what terms…, It is not sufficient to publish a review article in this reputed journal with the history, composition, contamination from different sources, ….as already published many reviews earlier.

I’m not really interested to read the plant list with phytochemistry, uses.., I’m much more interested that you provide an expert narrative indicating to the reader where you think there is promise and, conversely where there is little substance to claims.

Authors need to revise this MS, with discussion critically with some of the discussed point below.

1.       In the introduction explain the scientific gap why this kind of study is necessary with special reference to antimicrobials uses. What is the research gap and justify your aim of the review. How authors bring the special focus on animal husbandry

2.       Methodology needs to be clearly described how authors did their review work.

3.       Improve all the sections with interpretation by comparison of previous study.

4.       Table 1, figure 1, 2 not necessary as no scientific value added rather authors need to focus comparative study table(s) and importance of figures with scientific manner.

5.       Conclusion needs to be concise with the novel elements, not like a philosophical summary.

6.       My other concern is the authors are not updating with the recently published papers in past five years. These are the papers must be included-

[Study on odor control using wood vinegars (II). Application of wood vinegars to piggery wastes] - PubMed (nih.gov)

Study on the preparation of wood vinegar from biomass residues by carbonization process - ScienceDirect

Wood vinegar enhances humic acid-based remediation material to solidify Pb(II) for metal-contaminated soil - PubMed (nih.gov)

Comments on the Quality of English Language

Typological errors 

Author Response

(The authors gave the same response as above.)

Reviewer 3 Report

Comments and Suggestions for Authors

This paper reviews the environmental impacts caused by using antimicrobials in animal husbandry and health treatment and their implications for human health, as well as the use of wood vinegar as an alternative to antimicrobials. This manuscript is informative and interesting but needs to be improved and professional English editing to be published.

1.In the connection between the content of antibiotics and wood vinegar, some descriptions of why wood vinegar can replace antibiotics should be added as a transition.

2.There are many alternatives to antibiotics in the present study. Why is wood vinegar selected in this paper?

3.The article abstract mentioned: demand refining before further use in animal husbandry. Whether it means that wood vinegar can not be applied to animal husbandry in the current pollutants, how to carry out extensive application of wood vinegar and the feasibility of practical production and application.

4.There are too many keywords, which should be appropriately deleted according to the importance of the article.

5.It is suggested that a schematic diagram should be added as an aid to improve readers ' interest and understanding.

6.In the Wood Vinegar—Composition, Properties, and Uses part, the content of each part is not a parallel relationship, and needs to be re-summarized. The application in animals should be used as a separate part.

7.As a review article, it is not recommended to describe the discussion section separately, and it should be integrated into the corresponding content section.

8. Conclusions should be concise and forward-looking.

9. References do not match the citation format within the article. And pay attention to page numbers and periods in the reference. A large number of references lack key information, such as whether the journal is abbreviated or not, and inconsistent formatting.

10. “Environmental impact of antimicrobials” is an important part of the description in the paper and is better reflected in the title.

11. Authors should use tables to summarize the evidence for the use of wood vinegar in cattle, pigs, and poultry (including necessary information on the function of wood vinegar and related references). In addition, the potential mechanism of wood vinegar in the carbonization process and its potential to prevent environmental pollution caused by antibacterial agents need to be introduced and discussed.

12. Lack of author contributions and conflict of interest statements

Comments on the Quality of English Language

This manuscript requires professional English editing for publication.

Author Response

(The authors gave the same response as above.)

Reviewer 4 Report

Comments and Suggestions for Authors

The paper entitled “Wood Vinegar from the Carbonization Process and Its Potential to Prevent Environmental Contamination Caused by Antimicrobials from Animal Husbandry – A Review” is well-written and in a lucid manner. It is a very good attempt to apply deep learning for alternative natural products to replace conventional antimicrobials, highlighting wood vinegar (WV), also known as pyroligneous acid, as a promising product to fulfill this purpose. The manuscript can be accepted for publication in the journal ‘Animals’ by incorporating the following comments:

Line 30:  add an explanation of the abbreviation ..wood vinegar (WV)

Lines 53 – 58 and lines 96 -97: I recommend adding more recent citations:

https://doi.org/10.3390/ani12040470

https://www.mdpi.com/2077-0472/10/6/245

Lines 69 – 82: I recommend add table 1 in this section.

Lines 298 – 308:  I recommend adding more recent citation because not only milk but from cheese and milk products can be transferred antimicrobials residues: https://doi.org/10.3390/antibiotics10050570

Author Response

(The authors gave the same response as above.)

Reviewer 5 Report

Comments and Suggestions for Authors

Overall, the manuscript provides valuable information on previous studies using wood vinegar in animal husbandry systems and their detection and effects in soils. In addition, the authors discuss the limitations of using WV as an antimicrobial and provide suggestions to overcome those barriers. However, the article's English grammar need extensive revision. As a reviewer, I cannot provide all of the details for English corrections. In general, the authors use inflammatory and subjective language that is not appropriate for scientific publications such as "great concern", "basically", "considerably superior", "severe influence", "went far beyond", "indiscriminate use". All of these can be found in the introduction. Others include "baby hamster", hit-and-miss", and "so-called".

The authors use passive voice excessively, which is distracting to readers. For example, (lines 627-628) "In Table 1, the main compounds annotated in the chemical composition of eucalyptus WV and pointed out as having biological effects are named". First, I believe the authors meant to say "WV are pointed out", not "and". Second, this passive voice makes it difficult for the reader to understand. Instead, consider "Table 1 lists the main compounds in eucalyptus WV conferring biological effects". 

References need to be ordered in the number in which they appear in the text. Please refer to Animals guide for authors.

Title: Consider changing the title to "The potential of wood vinegar to prevent environmental contamination caused by antimicrobials used in animal husbandry - a review"

There is considerable repetitiveness in the article that discuss similar topics in different sections. For example, section 2.3 line 279-280 states "the environmental consequences....are the same as those of soil and water...." Because soil and water were already extensively discussed, there does not need to be a section repeating the same information. I recommend incorporating section 2.3 into the soil and water sections. Please review and make the manuscript more concise and organized. 

Line 99-100: Yes, antimicrobial (growth promoters) used in animal husbandry is used as a prophylactic (preventative) rather than to treat an infection (as in humans). Please clarify this. 

Line 103-104: Do the authors mean per kg of total animal weight or meat weight?

Lines 123-125: What are the manufacturing points? This sentence is vague.

Line 111-113: Is manure and urine used in irrigation practices? Please clarify.

Lines 127-137: This paragraph is important and discusses the actual impact of AMR on soil systems. 

Lines 138-140: how is manure/waste treated? What treatments do the authors refer to?

Line 157-159: Again, here is a repetition of the use of animal waste as fertilizer and in irrigation.  There are many repeating statements in this paper that need to be addressed through better organization.

Lines 168 and 173: It is "selective" pressure, not "selection" pressure

For section 2.3, there is valuable information that is more appropriate in the introduction. I recommend removing this section and incorporating relevant facts into the other sections.

Section 2.4 - This section is not relevant to the scope of the study, which is focused on reducing environmental contamination of antimicrobial residues. AMR residues in milk are not due to environmental contamination of AMR genes/bacteria. Meat contamination is not from the environment, but food processing facilities. Remove this section. 

Line 483: "Counts" should not be capitalized

Line 504: Use scientific name/latin name. This occurs throughout the article. Please ensure scientific names are used. 

Line 531: Check if this is how to properly cite this link

Line 536: Italicize S. Enteritidis

Line 541: S. Enteritidis is misspelled. 

Line 562: what does intestinal morphology refer to? Please clarify.

Line 564: Do the authors mean "omega"? Please use correct symbol. 

Line 565: There is an extra space between "acids. A concentration..."

Line 644: what does "cleanliness evolution" mean? Refining?

Line 649-650: Again, excessive use of passive voice. Consider revising to "Figure 2 displays one sample of refined eucalyptus and bamboo WV to illustrate the differences in color between WV types."

Line 668-670: This is very confusing to read. Please consult with someone to improve the English grammar. 

Line 693: Do not say "baby hamster". First, use latin name. Rephrase to "cells from X-day old latin name

Line 698: replace "by" with "but"

Line 706: Do the authors mean "asceptic"? It reads "asepsis"

Line 706-707: This sentence is confusing. What does EP mean?

Lines 763-767: This is not a scientific way to pose a question in a peer-reviewed publication. Please revise.

Lines 790-792: Is this not the same information as 1 and 2? Please avoid repetitiveness and be concise. 

Line 814-815 and 830: Here the authors list "tons", but then on 830 say "metric tons". Please be consistent with what units are listed in the paper. 

Line 851: Planted forests or "forest farms" are still damaging to biodiversity and deforestation does occur to create forest farms that are damaging to biodiversity. Please remove the statement "there is no contribution to deforestation". 

Comments on the Quality of English Language

This paper needs extensive English revision for grammar, overall organization, and sentence structure. In addition, there is considerable use on non-scientific terminology and phrases. 

Author Response

(The authors gave the same response as above.)
